## RESEARCH ARTICLE

# Low oxygen environment alters transcripts related to energy metabolism without altering the pluripotency core of bovine embryonic stem cells

Ramon C. Botigelli[§], Rachel Braz Arcanjo[‡], Carly Guiltinan[*], Amanda J. Morton, Justin M. Smith and Anna C. Denicol[‡]

## ABSTRACT

Stem cell pluripotency is shaped by culture microenvironment. Growth factors, cytokines, and oxygen tension are key regulators of self-renewal and pluripotency state. Here we evaluated the adaptation of bovine embryonic stem cells (bESC) derived on mouse embryonic fibroblasts (MEF) to feeder-free conditions and the effects of low oxygen (5% $O_2$) on their molecular profile. Primed bESC established on MEF and 21% $O_2$ (high oxygen) were adapted to 5% $O_2$ (low oxygen) and to feeder-free conditions in vitronectin to generate four culture conditions: high and low $O_2$ tension in MEF and feeder-free. Adaptation to feeder-free upregulated transcripts related to cell differentiation in bESC cultured in high compared to low oxygen. Transition of bESC on MEF from normoxia to 5% $O_2$ did not alter cell morphology or growth. Interestingly, colony morphology was maintained when transitioning bESC from MEF into feeder-free in low oxygen. Furthermore, there were fewer transcriptomic changes in bESC grown on MEF compared to feeder-free in low compared to high oxygen. While low oxygen tension did not alter the pluripotency profile of bESC, it promoted transcriptional changes related to energetic metabolism and increased mitochondrial membrane potential. These findings emphasize the importance of oxygen tension for derivation, maintenance, and performance of ESC.

KEY WORDS: Pluripotency, Oxygen, Metabolism, Transcriptome, Embryonic stem cell

## INTRODUCTION

Pluripotent stem cells (PSCs) can be obtained by deriving embryonic stem cells (ESCs) from the inner cell mass of preimplantation embryos or reprogramming differentiated cells, which are called induced pluripotent stem cells (iPSCs). Defining features of PSCs are self-renewal and the capability to differentiate into any cell type in the adult organism (Ying et al., 2008). Since the establishment of the first stable bovine ESC (bESCs) in 2018 (Bogliotti et al., 2018), livestock

PSCs have gained attention from the scientific community due to their numerous potential applications such as cellular agriculture, gamete generation (commercial interest and relevant model for human gametogenesis), production of gene-edited animals (to increase agricultural efficiency or model human conditions), and tissue generation for organ transplantation (Goszczynski et al., 2019a; Goszczynski et al., 2019b; Doncheva et al., 2021; Jara et al., 2023; Botigelli et al., 2023). With the increasing interest, recent studies have investigated ways to improve the derivation and maintenance of livestock ESCs. In this regard, various molecules, small molecules, and inhibitors have been used to promote the stable maintenance of livestock PSCs cultured on feeders or in feeder-free conditions (Zhao et al., 2021; Kinoshita et al., 2021; Zhi et al., 2024; Yang et al., 2025; Su et al., 2026).

Since the first report of ESC derivation from mouse blastocysts was published by Evans and Kaufman (1981), ESCs/iPSCs have been co-cultured with inactivated mouse embryonic fibroblasts (MEF). These embryonic somatic cells support the growth of PSCs by producing extracellular matrix (ECM) and fundamental factors such as Activin A and Leukemia Inhibitory Factor (LIF). Adapting PSCs to feeder-free conditions can be challenging, as it requires finding a suitable ECM (e.g. Matrigel, Geltrex, Vitronectin, Fibronectin, Laminin) and supplementing the factors typically provided by MEF. During this adaptation process, PSCs may experience cell death or differentiation due to the lack of support provided by MEF in feeder-free environments.

Past studies have emphasized the crucial role of the cellular micro and macroenvironment, including oxygen tension, in maintaining the pluripotency and differentiation potential of PSC. While atmospheric oxygen tension is about 21% (normoxia), the *in vivo* environment – particularly within the embryonic niche – is considerably lower, typically ranging from 1% to 5% (Simon and Keith, 2008). These physiological oxygen conditions have been used to enhance embryo quality during *in vitro* culture (Konstantogianni et al., 2024). Studies in mouse and human ESCs have demonstrated that hypoxia can increase expression of the core pluripotency factor POU5F1 and enhance the survival and proliferation of ESCs (Ezashi et al., 2005; Forristal et al., 2010). Hypoxia-inducible factors genes (HIFs) are key regulators of cellular responses to low oxygen levels, and act by modulating several pathways such as Notch, Wnt as well as canonical pluripotency markers that are involved in stem cell proliferation, self-renewal, response to stress and differentiation (Mazumdar et al., 2009; Forristal et al., 2010; Huang et al., 2018).

Initial reports of bESCs establishment cultured cells in normoxia (Bogliotti et al., 2018; Soto et al., 2021), but recent reports have pivoted toward maintenance of these cells in low oxygen tension (5% $O_2$; Kinoshita et al., 2021; Shirasawa et al., 2024; Yang et al., 2025; Zhi et al., 2024). However, the influence of oxygen tension on

Department of Animal Science, University of California, Davis, Davis, CA, 95616, USA.
*Present address: Department of Biomolecular Engineering, University of California Santa Cruz, Santa Cruz, CA, USA. ‡Present address: Departamento de Genética e Evolução, Universidade Federal de São Carlos, São Carlos, SP, Brazil.

§Authors for correspondence (rcbotigelli@ucdavis.edu; acdenicol@ucdavis.edu)

R.C.B., 0000-0002-2796-6062; R.B.A., 0000-0002-9758-2611; C.G., 0000-0002-4529-3818; A.C.D., 0000-0003-2528-4874

**Biology Open**

bESCs has not yet been systematically studied. Therefore, this work aimed to evaluate the impact of oxygen tension on the pluripotency and self-renewal of bESCs cultured on MEF or in feeder-free conditions. We first sought to compare bESC lines that were derived and maintained on MEF to vitronectin for feeder-free maintenance in normoxia. Then, we evaluated the adaptation of bESCs cultured on both MEF and vitronectin to low oxygen to assess whether there was a differential advantage for bESCs during this transition when maintained in an oxygen tension more suited to their physiology. Our results show that low oxygen mitigates molecular and transcriptional changes caused by transition of bESC from MEF to feeder-free conditions. This work contributes to a broader understanding of stem cell biology and provides evidence for the development of improved culture systems for basic research or practical applications in livestock biotechnology and agricultural production.

## RESULTS

### Transition of bESCs to feeder-free culture in normoxia disrupts colony formation but is not associated with transcriptomic changes to the pluripotency core

The bESC lines used in this study were derived on MEF and maintained for at least five passages on feeders in normoxia. Upon adaptation to feeder-free conditions, cells changed their morphology from flat colonies to monolayer-like growth, which has been described for cells cultured on vitronectin (VTN) in NBFR medium plus Activin A (Fig. 1A,C) (Guiltinan et al., 2025; Soto et al., 2021). Importantly, bESCs homogenously expressed the canonical pluripotent markers OCT4, SOX2, and NANOG in both feeder and feeder-free conditions (Fig. 1B,D). Moreover, when transitioned from feeder-free culture back to MEF, ESCs restored the colony morphology. To evaluate the changes associated with this transition, we examined the transcriptome of the same bESC lines cultured in MEF and after adaption to VTN culture. We identified a total of 402 upregulated and 459 downregulated genes for bESCs on VTN compared to bESCs on MEF (Fig. 2A). bESCs in both conditions (VTN and MEF) had similar expression of *POU5F1*, *SOX2*, *NANOG* and pluripotency markers differentially associated with pluripotent states (naive: *LIN28B*, *SALL4*, *KLF4*; primed: *OTX2*) (Fig. 2B, Fig. S1). We observed upregulation of *BMP1* and *ACVR2A* (TGF-β/SMADs pathway), *FZD9* (WNT/β-Catenin pathway), *TEAD3*, and *HDAC5* (epigenetic markers) on feeder-free bESCs (Fig. 2B). The upregulation of these genes could indicate mechanisms of adaptability of ESC through ECM remodeling. For instance, BMP1 cleaves ECM proteins like collagen and laminin, while TEAD3, a Hippo pathway effector, regulates proliferation and ECM sensing based on substrate changes. This profile also shows that a potential cell signaling compensation may occur with ACVR2A activating SMAD2/3 and FZD9 triggering canonical and non-canonical WNT pathways. HDAC5 can support pluripotency by repressing gene expression and regulating stress responses, alongside ACVR2A maintaining the primed state amid environmental shifts. Moreover, TEAD3 and BMP1 may predispose cells to lineage-specific differentiation under changing conditions. We observed *NODAL*, *JUN*, and *WNT7B* upregulated in the bESCs-MEF group (Fig. 2B). Those results highlighted the importance of the core of primed pluripotency maintenance, where NODAL sustains SMAD2/3 signaling (core to primed state) and JUN reinforces FGF/ ERK-driven primed identity. Moreover, NODAL and WNT7B could predispose cells derived in NBFR to mesoderm or neural fates if the pluripotency core were to be dysregulated. Interestingly, gene set enrichment analysis (GSEA) showed that the top upregulated biological processes of bESCs in the feeder-free condition were related to cell differentiation (e.g. mesenchyme development, skeletal

system morphogenesis, bone morphogenesis, developmental process, epithelial cell differentiation and others), while those in MEF had positive regulation of cell communication and regulation of signaling. Full GSEA results for enriched biological processes are shown in Table S1. Altogether, our analysis suggests that the canonical pluripotent markers were not affected by transitioning bESCs from MEF into feeder-free conditions under normoxia despite clear changes in cell morphology and colony formation ability. Nevertheless, the transcriptome/molecular profile of these cells indicated a shift suggesting preparation for cell differentiation.

### Adaptation to low oxygen environment preserves bESC identity when moved from MEF to feeder-free conditions

To investigate how low oxygen conditions could affect the pluripotency state of bESCs, we transferred bESCs cultured on MEF into low oxygen conditions. No detectable changes to cell morphology or growth were observed after adaptation to low oxygen. Then, adapted bESCs were transitioned into feeder-free conditions. We observed that under a low oxygen environment, bESC were able to maintain their morphology as colonies or clumps when transitioned to feeder-free culture. This colony shape resembled that of colonies observed on feeder layers, featuring defined borders, unlike the monolayer-like growth that occurred in feeder-free culture under high oxygen conditions (Fig. 3A,C). Interestingly, we tested a non-enzymatic cell dissociation reagent (ReLeSR) to passage clumps of bESCs cultured feeder-free in both low and high oxygen environments, aiming to promote their growth as colonies. We found that only the bESCs in low oxygen were capable of preserving this colony morphology. In contrast, clumps of bESCs cultured feeder-free in high oxygen did not exhibit the high density (packed colony-like) observed in low oxygen conditions and instead displayed a shape more similar to monolayer-like growth (Fig. S2).

Bovine ESCs in low oxygen expressed the canonical pluripotent markers *OCT4*, *SOX2*, and *NANOG* at similar levels regardless of culture matrix (Fig. 3B,D). To further explore the effects of low oxygen on bESCs, we compared their transcriptome when in MEF or VTN. We identified a total of 95 upregulated and 18 downregulated genes for bESCs in VTN compared to those on MEF cultured in low oxygen. Of note, the number of differentially expressed genes in bESCs cultured in low oxygen was smaller compared with the number observed in the previous experiment. Additionally, no differences were observed in the pluripotent core or important pathways such as JAK/STAT3, TGFb/SMADs, FGF/ERK, and WNT/b-CATENIN (Fig. S3). Interestingly, 28 genes (*FN1*, *TPM1*, *LOC112442408*, *PICALM*, *CDH11*, *TPM2*, *PPP3CB*, *EXT1*, *RNF11*, *FHL1*, *MBNL1*, *ZEB2*, *MBNL2*, *SAMD4A*, *HS6ST2*, *RAP2C*, *ATF5*, *LOC112442278*, *PRKG1*, *LOC101904796*, *RPS6KB1*, *CAMK2D*, *SIDT2*, *PRELP*, *TOX2*, *CD248*, *STRA6*, and *APOA1*) were similarly upregulated in feeder-free bESCs cultured in normoxia and low oxygen tension (Fig. 3E). Based on the limited number of DEGs genes, no enriched pathways were observed for GSEA analysis in low the $O_2$ comparison.

### Low oxygen environment promotes transcriptional changes related to energy metabolism and mitochondrial potential of bESCs

To elucidate the mechanisms and pathways affected by culture in low oxygen, we evaluated the transcriptome of bESCs cultured in feeder-free conditions in low oxygen versus high oxygen. We identified a total of 196 upregulated and 29 downregulated genes for bESCs in low oxygen compared to bESCs in high oxygen (Fig. 4A). No differences were observed on the expression of the core

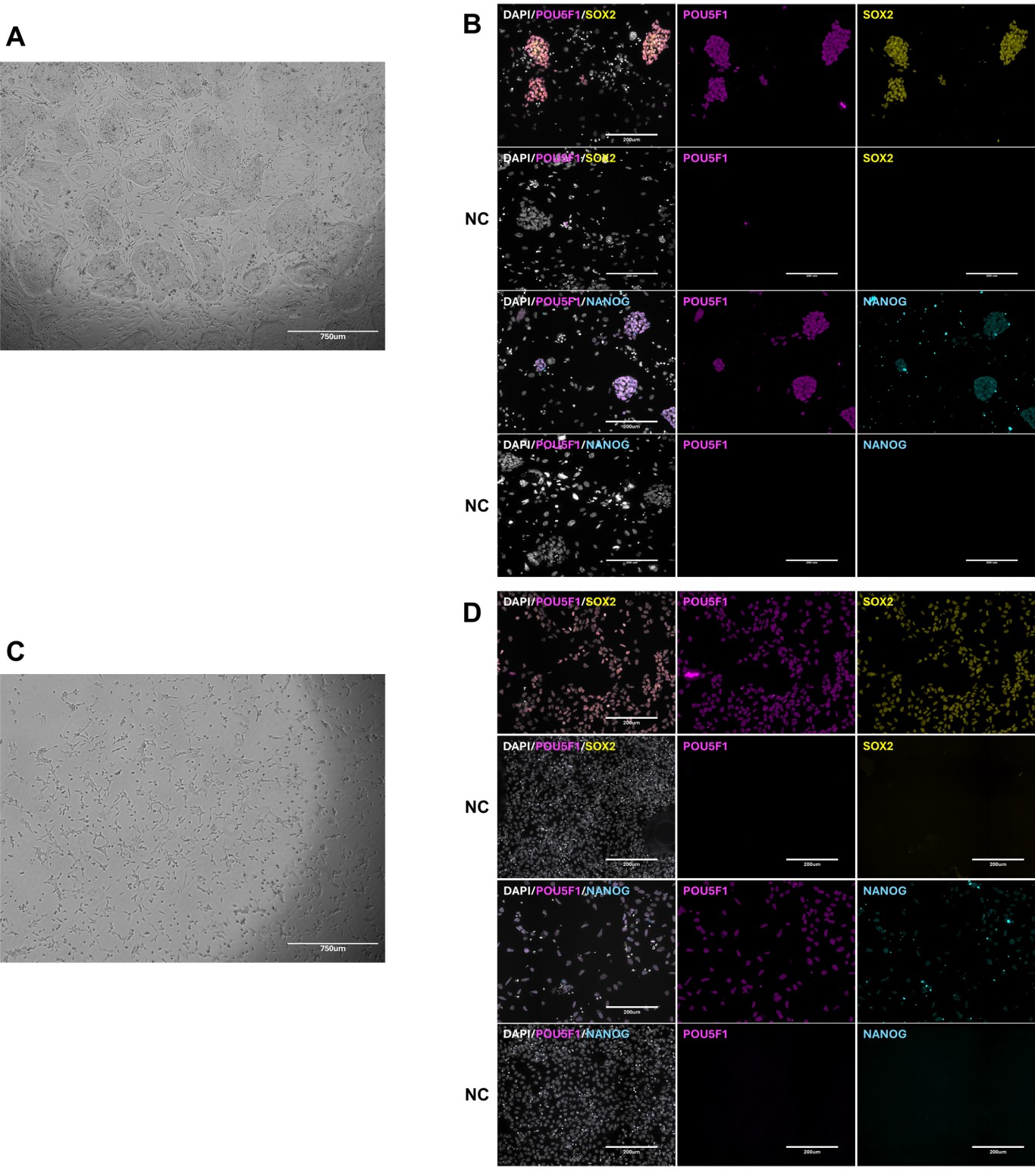

**Fig. 1. Characterization of bESCs cultured on MEF or feeder-free conditions in normoxia.** (A) Representative image of bESCs on MEF in normoxia. Scale bar: 750 µm. (B) Representative immunofluorescence images of bESCs on MEF in normoxia. NC, negative controls (primary antibody omitted). Scale bars: 200 µm. (C) Representative image of bESCs in feeder-free (vitronectin) in normoxia. Scale bar: 750 µm. (D) Representative immunofluorescence images of bESCs in feeder-free in normoxia. NC, negative controls (primary antibody omitted). Scale bars: 200 µm. In B and D, magenta indicates expression of POU5F1; yellow indicates expression of SOX2; cyan indicates expression of NANOG; gray indicates DNA. *N*=4 independent cell lines.

pluripotency markers *POU5F1*, *SOX2*, *NANOG*, *LIN28B*, *DNMT3B*, *SALL4*, *KLF4*, *MYC*, *STAT3*, and *OTX2* (Fig. S4A). Likewise, we did not observe differences on JAK/STAT3 and FGF/ ERK pathways. On the other hand, *NODAL* and *WNT5B*, important players of TGF-β/SMADs and WNT/β-Catenin pathways, were upregulated in low oxygen bESC (Fig. 4B). Additionally, many

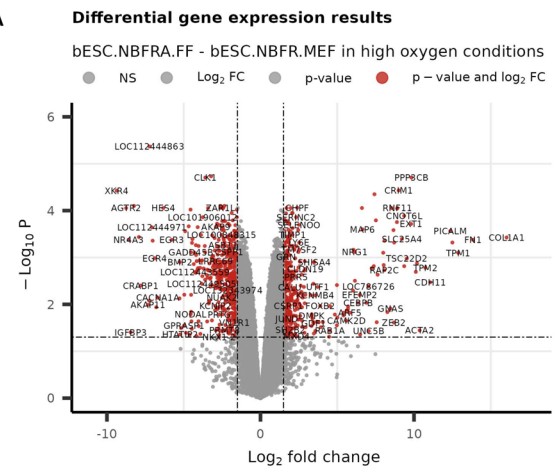

**Fig. 2.** See next page for legend.

**Fig. 2. Transcriptome comparison between bESCs cultured on MEF or feeder-free conditions under normoxia.** (A) Volcano plot showing significantly downregulated and upregulated genes between bESCs in feeder-free (FF; vitronectin) versus feeder (MEF) conditions cultured in high oxygen. (B) Heatmap of genes associated with JAK/STAT3, TGF-β/SMADs, FGF/ERK, and WNT/β-Catenin pathways and epigenetic markers for bESCs on MEF or feeder-free conditions cultured in high oxygen. Circles on the right side of gene indicate the presence of that gene in the DGE list. Brown circles show upregulated genes in bESC cultured in feeder (MEF). Natural sand-colored circles show upregulated genes in bESC cultured in feeder-free (FF). *N*=4 independent cell lines.

upregulated genes in low oxygen cultured cells were associated with oxidative phosphorylation and glycolysis (Fig. 4B). Corroborating that, over-representation analysis (ORA) of upregulated genes from bESCs in low oxygen showed biological processes related to metabolism, translation and response to hypoxia, while those in high oxygen had biological processes related to cell differentiation, cell migration and cellular response to multiple substances. Full ORA results for enriched biological processes are shown in Table S2. The top two enriched pathways of cells cultured in low oxygen tension were ribosome and oxidative phosphorylation (Fig. S4B,C). We also evaluated the effect of low oxygen on the protein expression of the pluripotency marker POU5F1. We did not observe differences in the normalized levels of POU5F1 on bESCs adapted in low oxygen compared with high oxygen (Fig. 4C,D). Given the importance of mitochondrial energy production for stem cell function, we next assessed whether these transcriptional changes could be confirmed by changes in mitochondrial membrane potential using TMRM staining followed by flow cytometry. Cells cultured in low oxygen exhibited increased levels of TMRM compared to those in high oxygen conditions, indicating a rise in mitochondrial membrane potential (Fig. 4E).

## DISCUSSION

Since the first report of stable maintenance of bovine ESC in 2018 (Bogliotti et al., 2018), research groups have slightly differed in their approaches to establish ESC lines from bovine embryos (Kinoshita et al., 2021; Soto et al., 2021; Zhao et al., 2021; Li et al., 2023; Shirasawa et al., 2024; Guiltinan et al., 2025). Most of them used FGF2 and WNT inhibitors (IWR-1 or XAV939) without or with Activin A (depending on the use of feeders or feeder-free conditions, respectively), with N2B27 or mTeSR serving as a base medium. Moreover, bESCs in those reports were all characterized as being in a primed state of pluripotency. So far, there have been no reports describing truly naïve bESCs, which indicates that there is still a gap of knowledge about PSCs in livestock species to recapitulate that state. To fill that gap, we investigated the effects of the transition of bESCs from culture on MEF into feeder-free conditions and the influence of low oxygen environment on the behavior and transcriptome of bESCs. Our findings show that bESCs can be adapted from MEF into feeder-free conditions in normoxia maintaining the expression of the pluripotent core, as previously reported (Soto et al., 2021; Guiltinan et al., 2025). The transcriptomic changes observed after this transition highlight the importance of growth factors and extracellular matrix provided by MEF. In addition to transcriptomic changes, we observed that oxygen tension significantly influenced the morphological behavior of bESCs following feeder withdrawal. Specifically, bESCs cultured in low oxygen tension maintained a colony or clump-like morphology, closely resembling the compact colonies observed on MEF feeder layers, with well-defined borders. This suggests that hypoxic conditions may partially compensate for the absence of

feeder cells by supporting cell–cell adhesion and colony integrity. The same behavior, or clump-like morphology was observed in bESC cultured in AFX medium using fibronectin and laminin as coating strategy under low oxygen conditions (Kinoshita et al., 2021).

Previous studies have demonstrated that a low oxygen environment promotes preimplantation embryo development in cows and humans (Herbemont et al., 2021; Konstantogianni et al., 2024; Boskovic et al., 2025) and the maintenance of mouse pluripotent stem cells by reducing spontaneous differentiation (Ezashi et al., 2005). Bovine ESCs have recently been established and maintained in hypoxic conditions (Kinoshita et al., 2021; Shirasawa et al., 2024; Yang et al., 2025; Zhi et al., 2024). Interestingly, the association of low oxygen environment with a different combination of small molecules, cytokines and growth factors in the medium was described as a model to establish formative bESCs (Yang et al., 2025; Zhi et al., 2024). Low oxygen was linked to shaping the epigenetic landscape during the early stages of bovine embryo development by changing DNA methylation and histone modification profiles (Gaspar et al., 2015; Bomfim et al., 2017; Cruz et al., 2023 preprint). In the current experiments, we observed that culture in low oxygen tension attenuated the changes in the transcriptomic profile when bESCs were transitioned from MEF into VTN. Furthermore, we did not observe differences in gene expression of epigenetic markers comparing bESCs in low oxygen versus high oxygen; however, epigenetic landscape changes have not been directly investigated, which warrants future studies.

Interestingly, fibronectin (*FN1*) was upregulated in feeder-free conditions in high and low oxygen (these experiments used vitronectin-coated plates), indicating that bESCs can produce an endogenous source of ECM to support their maintenance. The use of fibronectin was reported for the replacement of MEF in bESCs culture (Kinoshita et al., 2021; Zhao et al., 2021), where bESCs had colony shape in feeder-free conditions. Moreover, EDSCs (embryonic disc stem cells) described by Kinoshita et al. (2021) were cultured in medium containing 12.5 ng/ml of FGF2 and EPSC (expanded potential stem cells) described by Zhao et al. (2021) were cultured without FGF2 (but having LIF and Activin A). Our culture medium includes 20 ng/ml FGF2; we hypothesize that those differences in FGF2 concentrations may have an influence on the ability of bESCs to grow as colonies and not as monolayer-like growth. Additionally, our results may bring extra information regarding the optimal/preferred ECM by bESCs in feeder-free culture.

Mammalian cells produce ATP by adjusting the balance between glycolysis and oxidative phosphorylation (OXPHOS). PSCs have high metabolic demands to support their rapid division (Dahan et al., 2019). Although glycolysis yields significantly less ATP per glucose molecule compared to OXPHOS, the elevated glycolysis activity observed in primed PSCs offers some key advantages. Glycolysis can provide the necessary resources for the biosynthesis of nucleotides, lipids, and reducing equivalents, all of which are crucial for rapid cell replication. Also, reducing reactive oxygen species (ROS) generated by OXPHOS, it minimizes damage to both genomic and mitochondrial DNA, as well as reducing protein and lipid oxidation (Zhang et al., 2018; Dahan et al., 2019; Wang et al., 2023). In our low oxygen conditions, we observed an upregulation of many transcripts (*BNIP3*, *PDK1*, *SLC2A3*, *HK2*, *NDUFB3*, *COX17*, *NDUFC1*, *TPI1*, *NDUFA1*, *COA6*, *COX7A2*, *NDUFB1*, *NDUFA3*, *COX6A1*, and *COX6C*) involved in key steps of glycolysis, oxidative phosphorylation, and mitochondrial function. Our findings agree with the influence of hypoxic and ultrahypoxic (2% $O_2$) conditions during *in vitro* culture of bovine preimplantation embryos, where

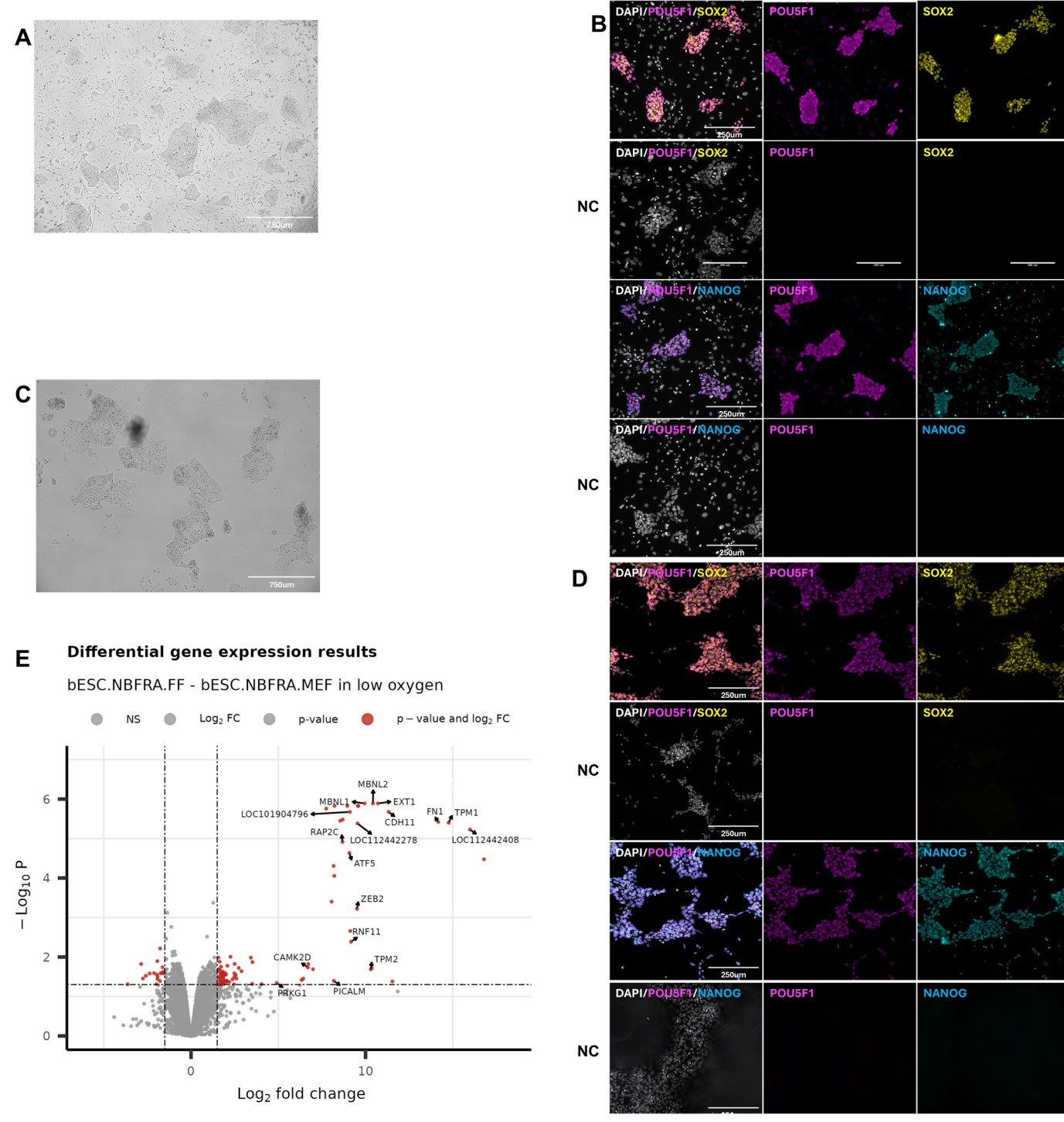

**Fig. 3. Characterization of bESCs cultured on MEF or feeder-free conditions after transition to low oxygen tension.** (A) Representative image of bESCs in feeder (MEF) in low oxygen tension. Scale bar: 750 μm. (B) Representative immunofluorescence images of bESCs on MEF in low oxygen. NC, negative controls (primary antibody omitted). Scale bars: 250 μm. (C) Representative image of bESCs in feeder-free (vitronectin) in low oxygen tension. Scale bar: 750 μm. (D) Representative immunofluorescence images of bESCs in feeder-free in low oxygen. NC, negative controls (primary antibody omitted). Scale bars: 250 μm. (E) Volcano plot showing significantly downregulated and upregulated genes between bESCs feeder-free versus feeder conditions cultured in low oxygen. Common upregulated genes observed in DGE list after bESC transition into feeder-free conditions (cultured under high oxygen and low oxygen) were highlighted. In B and D, magenta indicates expression of POU5F1; yellow indicates expression of SOX2; cyan indicates expression of NANOG; gray indicates DNA. *N*=4 independent cell lines.

genes involved in glycolysis and lipid metabolism were upregulated compared with normoxia (20% $O_2$; Boskovic et al., 2025). Moreover, these results may suggest that bESC in low oxygen environments increase their energy production using both mechanisms/pathways

to support proliferation and maintenance. Importantly, the increased mitochondrial membrane potential observed by TMRM analysis corroborates the transcriptomic findings. Lastly, we did not observe changes on the protein levels of POU5F1 in bESC feeder-free

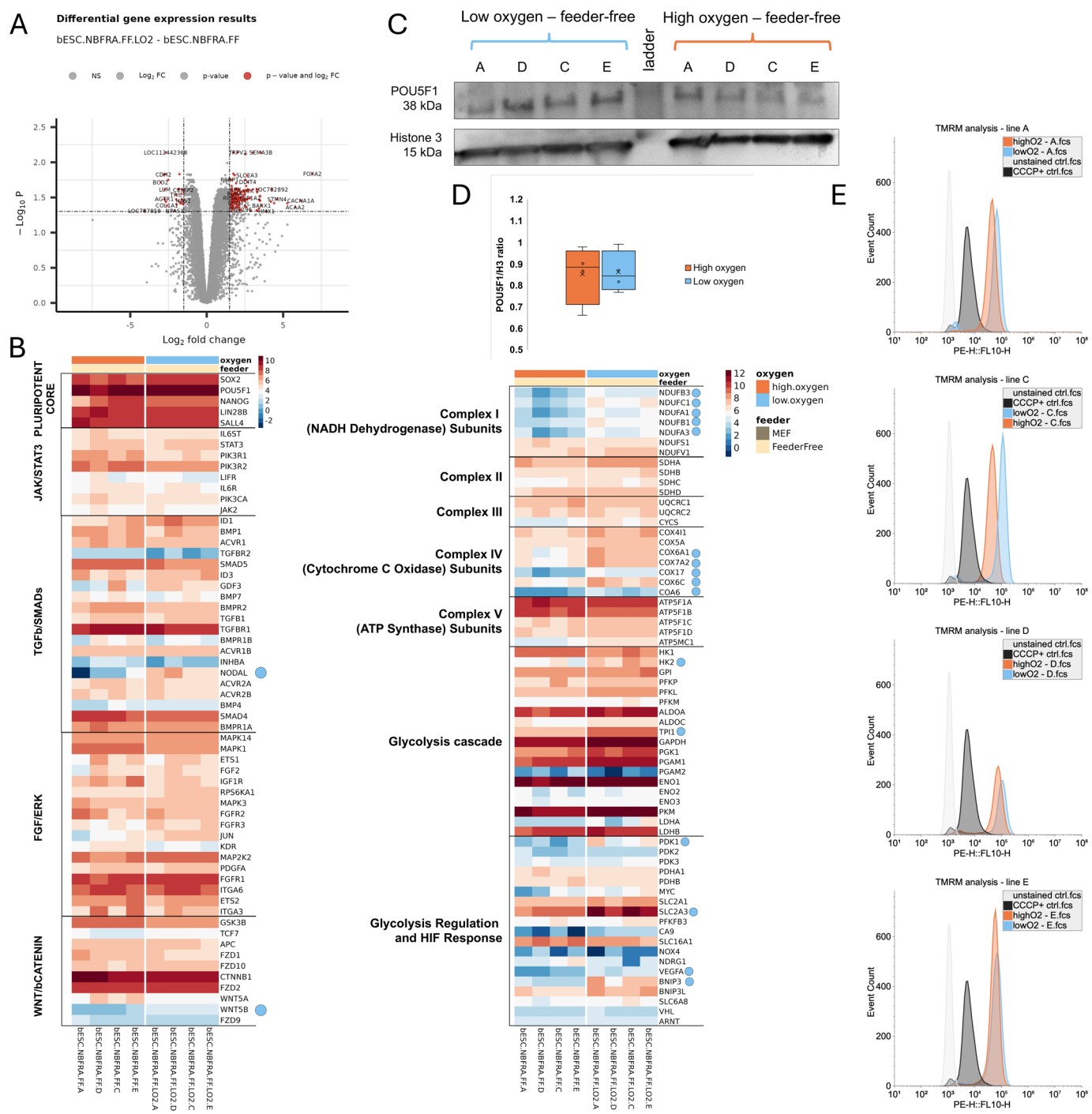

**Fig. 4. Comparison between bESCs in low versus high oxygen tension in feeder-free conditions.** (A) Volcano plot showing significantly downregulated and upregulated genes between bESCs in low versus high oxygen in feeder-free conditions. (B) Heatmap of genes associated with pluripotent core, JAK/STAT3, TGF-β/SMADs, FGF/ERK, and WNT/β-Catenin pathways and epigenetic markers for bESCs in low versus high oxygen in feeder-free conditions. Blue circles show upregulated genes in bESC cultured in feeder-free (FF) under low oxygen tension. (C) Western blot analysis of POU5F1 and Histone 3 proteins in bESCs cultured in low or high oxygen tension. (D) Plot showing quantification based on western blot of POU5F1 relative to Histone 3 in bESC lines cultured in low or high oxygen tension. N=4 independent cell lines. (E) Histogram representing bESCs in low versus high oxygen in feeder-free conditions after TMRM stain to evaluate mitochondrial membrane potential. N=two independent technical replicates using four cell lines per treatment. Note the clear shift in mitochondrial membrane potential in three out of four cell lines, with individual variation.

cultured in low compared to high oxygen tension, as previously reported on mESC in low oxygen conditions (Ezashi et al., 2005).

In conclusion, we have found that the transition from normoxia to low oxygen tension affects the energy metabolism of bESCs, but not the pluripotency core. Our findings bring valuable information

about the profile of bESCs on MEF and feeder-free conditions in normoxia and low oxygen tension and will be useful to optimize culture conditions and better define metabolic preferences of bESCs. Moreover, this work provides novel opportunities and information for comparative analyses that will contribute to the

establishment of bESCs with alternative features for targeted applications.

## MATERIALS AND METHODS
All experiments were performed in accordance with relevant guidelines and regulations.

### Culture of bESCs
All cell lines were derived from whole *in vitro*-produced bovine embryos under normoxia and on MEF (irradiated CF1 Mouse Embryonic Fibroblasts, Gibco #A34181) in NBFR medium as previously described (Guiltinan et al., 2025). NBFR medium consisted of 1:1 DMEM/F12 medium and neurobasal medium, 0.5% (v/v) N-2 supplement, 1% (v/v) B-27 supplement, 2 mM MEM non-essential amino acid solution, 1% (v/v) GlutaMAX supplement, 0.1 mM 2-mercaptoethanol, 1% (v/v) penicillin-streptomycin supplemented with 20 ng/ml human FGF2 and 2.5 µM IWR-1 at 37°C and 5% $CO_2$ and normoxia (Soto et al., 2021). For feeder-free culture, medium was supplemented with 20 ng/ml Activin A. The bESCs were harvested using Tryple express or ReleSR dissociation reagent and passaged every 3-5 days at a ratio of 1:4. All cell lines used in the present experiments undergo routine testing and are free of *Mycoplasma* sp. and other sources of contamination.

### Adaptation of NBFR-bESCs to low oxygen and feeder-free conditions
NBFR-bESCs passage 5 or 6 (four independent cell lines, each considered one biological replicate) were passaged into fresh MEF at a 1:5-1:10 split ratio in normoxia. After 24 h of culture, one plate was transferred to a low oxygen tension incubator (37°C, 5% $CO_2$, 5% $O_2$), while the other was maintained at normoxia. Cells were passaged every 4-5 days at a 1:4-1:8 split ratio in the presence of 10 µM Rho Kinase inhibitor Y27632 (ROCKi) onto fresh MEF. Cells were considered as being adapted to low oxygen after five passages. Then, cells were transitioned to feeder-free culture on vitronectin (VTNN or VTNXF) at either low or high oxygen at a 1:4 split ratio in the presence of 10 µM ROCKi. Cells were considered adapted to feeder-free culture after 3-4 passages on VTN and were passaged every 4-5 days at a 1:4-1:8 split ratio. Once adapted, the same four cell lines were used for technical repetitions.

### Protein immunolocalization
All four bESC lines in normoxia or low oxygen (MEF and feeder-free) were grown to 60-80% confluence and fixed using fresh 4% paraformaldehyde for 10 min at room temperature. After fixation, cells were permeabilized with 0.5% Triton-X100, then rinsed with DPBS and 0.05% Tween 20 (wash solution) three times for 5 min each and blocked for 1 h with 5% BSA in DPBS and incubated with primary antibody overnight at 4°C in 5% BSA and 0.3% Triton-X100 in DPBS solution. The next day, cells were rinsed three times for 10 min each with wash solution, then secondary antibodies were incubated for 1 h at room temperature. Then, cells were rinsed three times for 10 min each; on the second wash, 10 µg/ml of Hoescht 33342 was included for 10 min. All antibodies and dilutions used are included in Table S3. Negative controls were obtained by omitting the primary antibody incubation (secondary-only control). Images were captured using a confocal microscope (ImageXpress® Micro Confocal, Molecular Devices, San Jose, CA, USA).

### RNA extraction and transcriptome sequencing
RNA extraction was performed using a RNeasy Mini Kit (Qiagen) using DNase treatment (Qiagen). RNA was analyzed using a 2100 Bioanalyzer (Agilent Technologies). Libraries with unique adaptor barcodes were multiplexed and sequenced on a NovaSeq 6000 (paired-end, 150 base pair reads). Sequencing depth was at least 50 million reads per sample.

### RNA sequencing analysis
The quality of datasets was assessed using the FastQC tool. Raw reads were adapter and quality trimmed using Trimmomatic. Reads were aligned to the cow genome (NCBI, ARS-UCD2.0) with STAR. Optical duplicate reads were filtered using Picard (http://broadinstitute.github.io/picard/). Samtools was used to filter out alignments with MAPQ <30. Count matrices were generated using the featureCounts tool. A pure MEF sample was sequenced

and any reads with alignment to the bovine genome were removed from sample count matrices to normalize raw counts for bESCs cultured on MEF (Guiltinan et al., 2025). Information regarding sequencing batch can be found in Table S4. Raw counts were converted into counts per million using EdgeR, low counts genes were removed and the TMM (i.e. trimmed mean of M-values) function was applied to standardize data. Because samples separated primarily by the environment of culture (regular oxygen versus low oxygen) rather than batch-of-origin, no additional batch correction was applied. EdgeR was used to analyze the differentially expressed genes (DEGs). *P*-values were adjusted for the false discovery rate. Genes considered to be DEGs had threshold values of adjusted *P*-value <0.05 and fold change ≥1.5. Heatmap plots were made using filtered, normalized, counts per million reads data that were log2 transformed (i.e. Log2 CPM). GO analysis of enriched biological processes based on DEGs was performed using clusterProfile and GAGE package in R.

### Western blot
Cells were grown to 60-80% confluence and collected, washed in PBS, transferred to a 1.5 ml tube, and flash-frozen in liquid nitrogen. Proteins were extracted by adding 500 µl of RIPA buffer containing 1x protease inhibitor and 10 mM DTT directly to the tube followed by vigorous pipetting. The protein extracts were kept on ice for 2 h to allow the proteins to solubilize. Next, Micro BCA Protein assay kit was used to determine protein concentration of samples following manufacture instructions. Them, 1 ug of total protein was used for electrophoresis, 4X Laemmli buffer (Bio-Rad, cat. #1610747) were mixed with the samples and boiled at 95-100°C to denature the proteins. Proteins were loaded in a 10% precast gel and run at a constant 100 V for 60 min. After electrophoresis, proteins were transferred to a PVDF membrane in the mini-trans blot cell system. The membranes were blocked for 1 h at room temperature (RT) using a 3% BSA solution in Tris-buffered saline with 0.1% Tween-20 detergent (TBS-T). The POU5F1 antibody or Histone 3 antibody was added directly to the blocking solution, and the membranes were incubated overnight at 4°C under agitation. The next day, the membranes were washed five times for 3 min in TBST solution and incubated for 1 h at RT with peroxidase AffiniPure donkey anti-goat secondary antibody or peroxidase donkey anti-rabbit secondary antibody. Membranes were washed five times for 3 min and proteins were detected through incubation with SuperSignal West Pico PLUS chemiluminescent substrate. All antibodies and dilutions used are included in Table S3. Images were taken in ChemiDoc MP imaging system (Bio-Rad), and the band intensity was quantified using the Image Lab 5.1 software.

### Evaluation of mitochondrial membrane potential by flow cytometry
Cells were grown to 60-80% confluence and collected, rinsed with PBS −/− and then incubated with 20 nM of MitoProbe™ TMRM (a mitochondrial staining dye that is dependent on membrane potential) (Thermo Fisher Scientific) solution in culture medium for 30 min at 37°C. Carbonyl cyanide 3-chlorophenylhydrazone (CCCP) was used as a positive control for the induction of mitochondrial membrane depolarization. Unstained bESCs were used as a negative control. Cells were filtered using strainer 0.30 µm and then analyzed using flow cytometer (Cytoflex, Beckman Coulter Life Sciences), and the data for the cell counts were plotted as PE-Texas Red fluorescence intensity.

### Acknowledgements
The authors wish to acknowledge the support of the UC Davis Comprehensive Cancer Center Flow Cytometry Shared Resource, supported by the National Cancer Institute of the National Institutes of Health under award number P30CA093373. The content is solely the responsibility of the authors and does not necessarily represent the official views of the National Institutes of Health. The authors thank Ashley Karajeh for technical assistance in flow cytometry assays.

### Competing interests
The authors declare no competing or financial interests.

### Author contributions
Conceptualization: R.C.B., A.C.D.; Data curation: R.C.B., R.B.A., C.G., A.C.D.; Formal analysis: R.C.B., A.C.D.; Funding acquisition: A.C.D.; Investigation: R.C.B.,

R.B.A., C.G., J.M.S.; Methodology: R.C.B., R.B.A., C.G., J.M.S., A.J.M., A.C.D.; Project administration: A.C.D.; Resources: A.C.D.; Supervision: A.C.D.; Validation: R.C.B., R.B.A., C.G.; Visualization: R.C.B., A.C.D.; Writing – original draft: R.C.B., A.C.D.; Writing – review & editing: R.C.B., R.B.A., C.G., J.M.S., A.J.M., A.C.D.

## Funding

This work was funded by Genus plc. Open Access funding provided by University of California Davis. Deposited in PMC for immediate release.

## Data and resource availability

Additional raw data can be supplied by the authors upon reasonable request. RNA sequencing datasets generated by this report have been deposited to the GEO accession: GSE310875. All relevant data and details of resources can be found within the article and its supplementary information.

## Peer review history

The peer review history is available online at https://journals.biologists.com/bio/lookup/doi/10.1242/bio.062352.reviewer-comments.pdf

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
