## [Peer Review File · Biology Open]

Low oxygen environment alters transcripts related to energy metabolism without altering the pluripotency core of bovine embryonic stem cells

Ramon C. Botigelli, Rachel Braz Arcanjo, Carly Gultinan, Justin M. Smith, Amanda J. Morton and Anna C. Denicol
10.1242/bio.062352

Editor: Alissa Armstrong

Review timeline

Original submission:	2 November 2025
Editorial decision:	5 November 2025
First revision received:	1 April 2026
Editorial decision:	8 April 2026
Second revision received:	14 April 2026
Accepted:	15 April 2026

Original submission

First decision letter

MS ID#: bio.062352

MS Title: Low oxygen environment alters transcripts related to energy metabolism without altering the pluripotency core of bovine embryonic stem cells

Authors: Ramon C. Botigelli, Rachel Braz Arcanjo, Carly Gultinan, Justin M. Smith, Amanda J. Morton and Anna C. Denicol

I have now reached a decision on the above manuscript.

The reviewer reports are shown at the bottom of this email.

As you will see, the reviewers raised several substantial criticisms that prevent me from accepting the paper at this stage.

They suggest, however, that a revised version might prove acceptable if you can address their concerns. If you think that you can deal satisfactorily with the criticisms on revision, I would be pleased to see a revised manuscript. Please pay special attention to comments about proper controls, description of replicates, detailed methods, quantifying data, and providing functional evidence/not over-concluding. We would return a significantly revised manuscript to the reviewers.

At this stage, we also ask you to ensure your manuscript complies with our formatting guidelines. Provided you can fully address the referees' comments, we are positive about publication of your paper (we accept over 95% of revision submissions) and therefore hope you won't mind any extra work involved in reformatting your manuscript at this point.

Please upload both a 'clean' version of your Word file, along with a highlighted version clearly showing where you have made changes in the revised manuscript. Please avoid using 'Track changes' in Word files as these are lost in PDF conversion.

I should be grateful if you would also provide a point-by-point response detailing how you have dealt with the points raised by the reviewers in the 'Response to Reviewers' box. Please attend to all of the reviewers' comments. If you do not agree with any of their criticisms or suggestions please explain clearly why this is so.

Reviewer 1

Comments for the author

The manuscript addresses an interesting question in livestock stem cell biology: how oxygen tension influences pluripotency and metabolic signatures in bovine ESCs, particularly during transition from feeder to feeder-free culture. The conclusions (hypoxia supports feeder-free culture by dampening differentiation-associated transcriptional drift and altering metabolic programs without perturbing core pluripotency) are plausible and consistent with broader PSC literature in other species. The work is incremental but useful for the community.

Point-by-point review:

1. Experimental quality

a. Does each figure have the proper controls?

Pluripotency immunostaining: OCT4/SOX2/NANOG staining shown; appears qualitative only. No negative controls displayed.

RNA-seq: Differential expression analysis well described; MEF contamination correction included. The sample sizes are not indicated: how many samples were sequenced, were these biological or technical replicates, were the samples all from one experiment or were the experiments conducted on different times. This is important to assess if the differences were potentially a 'one-time event' or consistent.

Western blot: Only OCT4; loading control OK.

b. Are experiments performed using appropriate methods that will answer the question (or test the hypothesis or support the observations) posed by the authors? Is the right tool used for the job?

Yes for transcriptional changes; no for functional conclusions about "performance" and "maintenance" of pluripotency. The claim that low O₂ "enhances maintenance" is not experimentally demonstrated beyond expression maintenance.

c. Were the data analysed using appropriate statistical tests?

Yes

2. Reproducibility

a. Were experiments in each figure performed using adequate number of biological replicates? 4 bESC lines initially mentioned, but per-figure replicate number is unclear, only a sample of the data is shown. The lines are not described: were these previously established in the lab, or newly for this project. What are the source embryos, how were these lines characterized as pluripotent? Figures do not show biological replicate dot plots or clustering; DEG tables imply replicates but not explicitly shown.

b. Is there sufficient raw data to assess the rigor of the analysis?

Sequencing data availability is mentioned but not deposited yet. The imaging is qualitative only, and only a sample of the stainings is shown (if this was indeed replicated on 4 lines)

c. Does the methods section provide sufficient detail to permit reproducibility?

Yes

3. Completeness

a. Are the author's conclusions supported by the data?

Partially

- The core pluripotency gene expression by staining is based on qualitative analysis and precludes statements such as 'homogeneously expressed'. The authors could quantify the images or change the wording. In the sequencing data, homogeneously expressed is also not correct: this is bulk RNA sequencing. The authors could say that they are consistently expressed across samples, if this is the case.
- The authors interpret changes in ECM-related genes as "adaptation", this is mechanistically plausible but not tested.
- There are no functional tests or replication/confirmation experiments
- The authors claim that hypoxia affects energy metabolism, but, while likely, this is solely based on gene-expression.

b. Are there any flaws in the experimental design that invalidate the approach taken by the authors?

None apparent. It is though important to clarify if these experiments were confirmed in any manner, or if this was all done in one experiment.

c. Are there experiments that have not been performed, but if true would disprove the conclusion? If yes, and if such experiments would be costly or time-consuming to perform, do the authors acknowledge this in a discussion of the limitations?

At this stage, it is unclear if the replication experiments were carried out. The data could be shown in more completeness, and the images could be quantified for fluorescent signal intensity per cell.

4. Scholarship

- a. Do the authors cite and discuss the merits of relevant data that would argue against their conclusion?
- b. Do the authors cite and discuss the merits of relevant data that would support their conclusion?
- c. For techniques/methods manuscripts, Do the authors cite and discuss the current state of the field and clearly explain how the method improves the field?

This appears correct

In summary, this manuscript provides incremental but valuable insight into how oxygen tension affects bovine ESC transcriptional profiles during transition from feeder-dependent to feeder-free conditions. The observations are broadly consistent with established pluripotent stem cell biology in other species and will be of interest to the livestock stem cell community. However, several aspects of data presentation and interpretation limit the strength of the conclusions in the current form. In particular, the claims regarding enhanced maintenance of pluripotency and metabolic adaptation under hypoxia rely primarily on qualitative imaging and gene-expression signatures, without functional validation or quantitative analysis. The number of biological replicates and whether the presented results represent a single experiment or multiple independent repeats should be clarified, and the authors should either include quantitative measures (e.g., image quantification, replicate-level RNA-seq data presentation) or temper their wording accordingly. With revisions to explicitly document replicates, strengthen (or soften) claims regarding pluripotency maintenance and metabolic effects, and ensure that qualitative observations are framed appropriately, the manuscript would meet the journal's standards. I encourage the authors to expand figure legends and methods to clarify replication, consider quantifying imaging where feasible, and revise interpretive statements to match the evidence presented. With these improvements, the work will provide a clearer and more rigorous contribution to the field.

Reviewer 2

Comments for the author

Bottigelli et al.'s manuscript describes work to assess the effect of oxygen conditions on the culture of bovine embryonic stem cells. This issue has not been systematically studied, although prior studies have used hypoxic culture conditions without explicit comparison with normoxic conditions. The study assesses pluripotency markers by immunofluorescence staining, examines cell morphology and performs transcriptomic analysis in cultures with/without feeder cells and in

hypoxia/normoxia. The manuscript is well-written. Given the relative recency of culture conditions for bovine ES cells, these descriptive data are likely to be of interest to the field.

Major comments:

- * There is a danger of overinterpretation of the RNA sequencing data without validation or functional evidence. I wasn't clear about why the authors think that the normoxic cells that transition to feeder-free conditions are "primed toward commitment to the next events of cell differentiation" given they express all of the core pluripotency factors. Moreover, the authors don't show that the cells synthesize more fibronectin protein as "an endogenous source of ECM to support their maintenance" or "rebuild their niche" through expression of CDH11 during the transition to feeder-free culture - these claims are speculative. Likewise, I'm not sure one can conclude that "a transition to low oxygen conditions boosted glycolysis" based on gene expression data alone so the discussion of this issue is also speculative.
- * The authors should clarify how they treat the mouse reads in RNA sequencing experiments in which bovine cells are cultured with mouse cells. I didn't understand the methods section which states that a pure MEF sample was used for "normalization".
- * Details of technical and experimental replicates should be included in Figure legends.

Minor comments:

- * The abstract begins with what the authors do in the study. It would benefit from a couple of introductory sentences with background and their motivation for the study.
- * The Boglioti paper should be cited in the introduction following "the establishment of the first stable bovine ESC (bESCs) in 2018".
- * The authors desire to share RNA sequencing data via GEO is a positive. These datasets could be deposited in GEO now to generate an accession number (and embargoed until publication if the authors wished).
- * What subline of "MEFs" was used in the study and where were they sourced from?
- * The discussion does not discuss the finding that colony morphology is affected by oxygen conditions following feeder cell withdrawal.
- * The descriptions of data in the Figure legends could be improved. What do the circles next to some genes mean (e.g. Figure 2B)?
- * The cells in this study were isolated on feeder cells. I would be interested in the authors opinions on how the different conditions would affect derivation. Would isolating the cells in hypoxia vs normoxia alter the nature of the cells amenable to culture, or the quality of the resulting culture?

Reviewer's Responses to Questions

Experimental quality

Does each figure have the proper controls?

If 'No', please indicate reasons in Comments for Author box below.

Reviewer #1:

- No

Reviewer #2:

- Yes

Were the data analyzed using appropriate statistical tests?

If 'No', please indicate reasons in Comments for Author box below.

Reviewer #1:

- Yes

Reviewer #2:

- Yes

Reproducibility

Were experiments performed using adequate number of biological replicates?

If 'No', please indicate reasons in Comments for Author box below.

Reviewer #1:

- No

Reviewer #2:

- No

Does the methods section provide sufficient detail to permit reproducibility?

If 'No', please indicate reasons in Comments for Author box below.

Reviewer #1:

- No

Reviewer #2:

- No

Completeness

Are the manuscript's conclusions supported by the data?

If 'No', please indicate reasons in Comments for Author box below.

Reviewer #1:

- No

Reviewer #2:

- No

Scholarship

Do the authors cite and discuss the merits of data that would argue for and against their conclusion?

If 'No', please indicate reasons in Comments for Author box below.

Reviewer #1:

- Yes

Reviewer #2:

- Yes

Does the manuscript title & abstract accurately reflect the contents of the manuscript, without hyperbole?

If 'No', please indicate reasons in Comments for Author box below.

Reviewer #1:

- Yes

Reviewer #2:

- Yes

First revision

Author response to reviewers' comments

Dear Editor and reviewers of Biology Open,

We would like to thank you for taking the time to provide a detailed review of our manuscript (bio.062352, entitled "Low oxygen environment alters transcripts related to energy metabolism without altering the pluripotency core of bovine embryonic stem cells" and to point out areas for

improvement. Please find our responses to each of your questions below. We hope we were able to address your concerns.

Reviewer 1: The manuscript addresses an interesting question in livestock stem cell biology: how oxygen tension influences pluripotency and metabolic signatures in bovine ESCs, particularly during transition from feeder to feeder-free culture. The conclusions (hypoxia supports feeder-free culture by dampening differentiation-associated transcriptional drift and altering metabolic programs without perturbing core pluripotency) are plausible and consistent with broader PSC literature in other species. The work is incremental but useful for the community.

1. Experimental quality

a. Does each figure have the proper controls?

Pluripotency immunostaining: OCT4/SOX2/NANOG staining shown; appears qualitative only. No negative controls displayed.

Authors: We have added a negative control sample (bESC) in all rounds of immunostaining. That negative control reaction was made following the methods described on Methods/ Protein immunolocalization but omitting the presence of primary antibody. We also included the description of the negative control reaction in that section.

RNA-seq: Differential expression analysis well described; MEF contamination correction included. The sample sizes are not indicated: how many samples were sequenced, were these biological or technical replicates, were the samples all from one experiment or were the experiments conducted on different times. This is important to assess if the differences were potentially a 'one-time event' or consistent.

Western blot: Only OCT4; loading control OK.

Authors: Thank you for bringing this to our attention and we apologize for lack of information regarding the biological or technical replicates in the methods section. We considered each bESC line as a biological replicate; four lines were sequenced. All sequencing experiments were conducted using the same bESC lines, after appropriated adaption in the described condition. Sample sequencing was performed in different moments. We included a table in the supplemental materials section to include all information regarding those samples. Before any comparison, raw counts were converted into counts per million using EdgeR package, low counts genes were removed and the TMM (i.e. Trimmed Mean of M-values) function was applied to standardize data and reduce potential batch effects.

b. Are experiments performed using appropriate methods that will answer the question (or test the hypothesis or support the observations) posed by the authors? Is the right tool used for the job?

Yes for transcriptional changes; no for functional conclusions about "performance" and "maintenance" of pluripotency. The claim that low O₂ "enhances maintenance" is not experimentally demonstrated beyond expression maintenance.

Authors: Thank you for bringing this to our attention. We have reworded the text to reflect that the cells demonstrated transcriptional changes.

2. Reproducibility

a. Were experiments in each figure performed using adequate number of biological replicates?

4 bESC lines initially mentioned, but per-figure replicate number is unclear, only a sample of the data is shown. The lines are not described: were these previously established in the lab, or newly for this project. What are the source embryos, how where these lines characterized as pluripotent? Figures do not show biological replicate dot plots or clustering; DEG tables imply replicates but not explicitly shown.

Authors: We used four bESC lines (biological replicates) derived from four bovine embryos in our lab in the same culture conditions (NBFR medium / on MEFs / high oxygen). All assays to demonstrate the pluripotency of those cell lines were published recently by our lab (Guiltinan et al 2025 - 10.1242/bio.061819). Here, we used those same cell lines to evaluate the effect of the low oxygen on bESC. Before and after low oxygen adaptation (MEFs and feeder-free), all four bESC lines were submitted to IF assays, and all demonstrated the same pattern of presence of the canonical pluripotent markers (POU5F1, SOX2, and NANOG) on those different conditions. We included only one image per treatment (high/low oxygen on MEFs/Feeder-free) to not be redundant. Our goal in that experiment was to highlight the presence of those canonical pluripotent markers, as a qualitative assay. For the heatmaps, we decided to include all samples instead of the average of

the group (Feeder-Free vs MEF or high oxygen vs low oxygen) to highlight the small individual changes across biological replicates.

b. Is there sufficient raw data to assess the rigor of the analysis?

Sequencing data availability is mentioned but not deposited yet. The imaging is qualitative only, and only a sample of the stainings is shown (if this was indeed replicated on 4 lines)

Authors: We addressed this concern in the previous question. We will rephrase sentences to highlight all four bESC lines were stained. We have clarified this within the methods (line 115).

c. Does the methods section provide sufficient detail to permit reproducibility?

Yes

3. Completeness

a. Are the author's conclusions supported by the data?

Partially

- The core pluripotency gene expression by staining is based on qualitative analysis and precludes statements such as 'homogeneously expressed'. The authors could quantify the images or change the wording. In the sequencing data, homogeneously expressed is also not correct: this is bulk RNA sequencing. The authors could say that they are consistently expressed across samples, if this is the case.

- The authors interpret changes in ECM-related genes as "adaptation", this is mechanistically plausible but not tested.

- There are no functional tests or replication/confirmation experiments

- The authors claim that hypoxia affects energy metabolism, but, while likely, this is solely based on gene-expression.

Authors: Sentences were removed or reworded to avoid a perception of speculation. In response to the reviewer's concern, we decided to perform a functional test of energy metabolism shift as we too consider that this is a very important point and should be confirmed. This experiment was added to the revised version of the manuscript.

b. Are there any flaws in the experimental design that invalidate the approach taken by the authors?

None apparent. It is though important to clarify if these experiments were confirmed in any manner, or if this was all done in one experiment.

Authors: We have clarified this within the methods (line 112). The adaption was made in one round. After that, the same four cell lines were used multiple times for all experiments and technical repetitions (IF), except bulk RNA-seq and WB (one repetition - counting cell line as biological repetition).

c. Are there experiments that have not been performed, but if true would disprove the conclusion? If yes, and if such experiments would be costly or time-consuming to perform, do the authors acknowledge this in a discussion of the limitations?

At this stage, it is unclear if the replication experiments were carried out. The data could be shown in more completeness, and the images could be quantified for fluorescent signal intensity per cell.

Authors: We have clarified this within the methods (line 103). We discussed the option to quantify the fluorescent signal intensity from IF images, however, we do not strongly believe that this would be an accurate way to compare pluripotency between samples. Therefore, we chose the classic pluripotency marker OCT4 for protein concentration quantification by WB.

4. Scholarship

a. Do the authors cite and discuss the merits of relevant data that would argue against their conclusion?

b. Do the authors cite and discuss the merits of relevant data that would support their conclusion?

c. For techniques/methods manuscripts, Do the authors cite and discuss the current state of the field and clearly explain how the method improves the field?

This appears correct

In summary, this manuscript provides incremental but valuable insight into how oxygen tension affects bovine ESC transcriptional profiles during transition from feeder-dependent to feeder-free conditions. The observations are broadly consistent with established pluripotent stem cell biology in other species and will be of interest to the livestock stem cell community. However, several aspects of data presentation and interpretation limit the strength of the conclusions in the current

form. In particular, the claims regarding enhanced maintenance of pluripotency and metabolic adaptation under hypoxia rely primarily on qualitative imaging and gene-expression signatures, without functional validation or quantitative analysis.

The number of biological replicates and whether the presented results represent a single experiment or multiple independent repeats should be clarified, and the authors should either include quantitative measures (e.g., image quantification, replicate-level RNA-seq data presentation) or temper their wording accordingly.

With revisions to explicitly document replicates, strengthen (or soften) claims regarding pluripotency maintenance and metabolic effects, and ensure that qualitative observations are framed appropriately, the manuscript would meet the journal's standards. I encourage the authors to expand figure legends and methods to clarify replication, consider quantifying imaging where feasible, and revise interpretive statements to match the evidence presented. With these improvements, the work will provide a clearer and more rigorous contribution to the field.

Authors: We have clarified throughout the manuscript how all analyses were performed, including the number of replicates and biological samples used. Regarding the quantitative analyses, we appreciate the reviewer's comment and would like to emphasize that these data are presented in Figure 4 (Western blot analysis comparing bESCs cultured under high versus low oxygen conditions). We consider this technique a gold standard for protein quantification. Additionally, we have incorporated a new functional assay to further support and highlight the observed changes in energy metabolism of bESCs under low oxygen conditions.

Reviewer 2: Bottigelli et al.'s manuscript describes work to assess the effect of oxygen conditions on the culture of bovine embryonic stem cells. This issue has not been systematically studied, although prior studies have used hypoxic culture conditions without explicit comparison with normoxic conditions. The study assesses pluripotency markers by immunofluorescence staining, examines cell morphology and performs transcriptomic analysis in cultures with/without feeder cells and in hypoxia/normoxia. The manuscript is well-written. Given the relative recency of culture conditions for bovine ES cells, these descriptive data are likely to be of interest to the field.

Major comments:

* There is a danger of overinterpretation of the RNA sequencing data without validation or functional evidence. I wasn't clear about why the authors think that the normoxic cells that transition to feeder-free conditions are "primed toward commitment to the next events of cell differentiation" given they express all of the core pluripotency factors. Moreover, the authors don't show that the cells synthesize more fibronectin protein as "an endogenous source of ECM to support their maintenance" or "rebuild their niche" through expression of CDH11 during the transition to feeder-free culture - these claims are speculative. Likewise, I'm not sure one can conclude that "a transition to low oxygen conditions boosted glycolysis" based on gene expression data alone so the discussion of this issue is also speculative.

Authors: We appreciate the comment. We have reworded the manuscript in several parts to avoid overinterpreting the transcriptomic data. Regarding the energy metabolism observations, we conducted an additional experiment to examine mitochondrial membrane potential in cells cultured under high and low O₂ tension. The results are now part of the revised manuscript.

* The authors should clarify how they treat the mouse reads in RNA sequencing experiments in which bovine cells are cultured with mouse cells. I didn't understand the methods section which states that a pure MEF sample was used for "normalization".

Authors: We added information to clarify this point.

* Details of technical and experimental replicates should be included in Figure legends.

Authors: This was done.

Minor comments:

* The abstract begins with what the authors do in the study. It would benefit from a couple of introductory sentences with background and their motivation for the study.

Authors: We have changed the abstract to include the information requested (lines 14-16).

* The Boglioti paper should be cited in the introduction following "the establishment of the first stable bovine ESC (bESCs) in 2018".

Authors: We have included the citation (line 43).

* The authors desire to share RNA sequencing data via GEO is a positive. These datasets could be deposited in GEO now to generate an accession number (and embargoed until publication if the authors wished).

Authors: We have tried to upload the RNA sequencing data before the first submission. However, NCBI/GEO wasn't accepting the submission during the government shutdown. The RNA sequencing data are now deposited under the GEO accession: GSE310875 and embargoed until publication. If necessary, access to the raw data can be obtained using the following token: qtmrokmqjzkjbir

* What subline of "MEFs" was used in the study and where were they sourced from?

Authors: We have included this information in the Methods section (line 96).

* The discussion does not discuss the finding that colony morphology is affected by oxygen conditions following feeder cell withdrawal.

Authors: We have included this point in the discussion.

* The descriptions of data in the Figure legends could be improved. What do the circles next to some genes mean (e.g. Figure 2B)?

Authors: The circle highlights genes that were present in the DGE list of that comparison. We have added it to the figure legend (line 43).

* The cells in this study were isolated on feeder cells. I would be interested in the authors opinions on how the different conditions would affect derivation. Would isolating the cells in hypoxia vs normoxia alter the nature of the cells amenable to culture, or the quality of the resulting culture?

Authors: To our knowledge, there are no studies that have systematically derived ESCs from the same species in parallel under high and low oxygen conditions to directly address this question. Therefore, it remains unclear whether oxygen tension during the derivation phase itself has an impact distinct from subsequent culture conditions. In the absence of such comparative studies, we can hypothesize that derivation under low oxygen conditions may yield ESC lines with characteristics similar to those we observed after adaptation to hypoxia, such as change in colony morphology, altered metabolic profiles, and potentially reduced divergence upon transition to feeder-free conditions. This hypothesis has not yet been experimentally tested and represents an important direction for future research.

Second decision letter

MS ID#: bio.062352R1

MS Title: Low oxygen environment alters transcripts related to energy metabolism without altering the pluripotency core of bovine embryonic stem cells

Authors: Ramon C. Botigelli, Rachel Braz Arcanjo, Carly Gultinan, Justin M. Smith, Amanda J. Morton and Anna C. Denicol

I have now reached a decision on the above manuscript.

The reviewer reports are shown at the bottom of this email.

As you will see, both reviewers appreciate how their comments and concerns were taken into consideration to improve the manuscript. Reviewer 2 raised some additional points that will require amendments to your manuscript. Upon satisfactory adjustment according to those comments, the paper will be suitable for publication.

At this stage, we also ask you to ensure your manuscript complies with our formatting guidelines - please see our manuscript preparation guidelines for details. Provided you are able to fully address the referees' comments, we are positive about publication of your paper (we accept over 95% of revision submissions) and therefore hope you won't mind any extra work involved in reformatting your manuscript at this point.

Please upload both a 'clean' version of your Word file, along with a highlighted version clearly showing where you have made changes in the revised manuscript. Please avoid using 'Track changes' in Word files as these are lost in PDF conversion.

I should be grateful if you would also provide a point-by-point response detailing how you have dealt with the points raised by the reviewers in the 'Response to Reviewers' box. Please attend to all of the reviewers' comments. If you do not agree with any of their criticisms or suggestions please explain clearly why this is so.

Reviewer 1

Comments for the author

I thank the authors for taking my comments into consideration and adjusting the manuscript to remove speculation and adhere to the findings supported by the data. The inclusion of the validation of metabolic changes in the lines strengthens the study.

Reviewer 2

Comments for the author

The manuscript is much improved and the authors have generally addressed my previous concerns well. I have a few remaining comments:

* In legends, the nature of replicates could still be clarified. For example, in Fig 4D, is the bar plot a simple quantification of the four lanes shown in C or does it include additional technical (blot) or experimental (protein lysate) replicates? Plotting individual datapoints here would also improve transparency.

* In the new Fig 4E, the legend asks readers to note the shift in 3/4 cell lines. Are these the only technical replicates for each sample? Can any statistics be performed to give confidence that there is reproducibly a shift?

* The MEF-contamination correction approach is unusual but pragmatic; it would benefit from a brief explicit statement of what exactly is subtracted or filtered rather than directing readers to Gultinan et al. (2025).

* Trimmed Mean of M-values is a library size correction, rather than batch correction. The authors should clarify whether a separate batch correction was performed?

* The GEO accession number was provided to reviewers but has not been updated in the manuscript itself.

* Some references seem to be incomplete (e.g. Yang et al. (2024) and Zhi et al. (2024) lack journal information).

Reviewer's Responses to Questions

Experimental quality

Does each figure have the proper controls?

If 'No', please indicate reasons in Comments for Author box below.

Reviewer #1:

- Yes

Reviewer #2:

- Yes

Were the data analyzed using appropriate statistical tests?

If 'No', please indicate reasons in Comments for Author box below.

Reviewer #1:

- Yes

Reviewer #2:

- Yes

Reproducibility

Were experiments performed using adequate number of biological replicates?

If 'No', please indicate reasons in Comments for Author box below.

Reviewer #1:

- Yes

Reviewer #2:

- No

Does the methods section provide sufficient detail to permit reproducibility?

If 'No', please indicate reasons in Comments for Author box below.

Reviewer #1:

- Yes

Reviewer #2:

- Yes

Completeness

Are the manuscript's conclusions supported by the data?

If 'No', please indicate reasons in Comments for Author box below.

Reviewer #1:

- Yes

Reviewer #2:

- Yes

Scholarship

Do the authors cite and discuss the merits of data that would argue for and against their conclusion?

If 'No', please indicate reasons in Comments for Author box below.

Reviewer #1:

- Yes

Reviewer #2:

- Yes

Does the manuscript title & abstract accurately reflect the contents of the manuscript, without hyperbole?

If 'No', please indicate reasons in Comments for Author box below.

Reviewer #1:

- Yes

Reviewer #2:

- Yes

Second revision

Author response to reviewers' comments

Dear Editor and reviewers of Biology Open,

We would like to thank you for taking the time to provide a detailed review of our manuscript (bio.062352 - revision 1) and to point out areas for improvement. Please find our responses to each of your questions below. We hope we were able to address your concerns.

Comments from the Reviewers:

Reviewer 1: I thank the authors for taking my comments into consideration and adjusting the manuscript to remove speculation and adhere to the findings supported by the data. The inclusion of the validation of metabolic changes in the lines strengthens the study.

Authors: We appreciate the recognition of our adjustments to improve quality and relevance of the work.

Reviewer 2: The manuscript is much improved and the authors have generally addressed my previous concerns well. I have a few remaining comments:

Authors: We appreciate the recognition of our changes. A detailed point-by-point response to reviewer comment is provided below.

* In legends, the nature of replicates could still be clarified. For example, in Fig 4D, is the bar plot a simple quantification of the four lanes shown in C or does it include additional technical (blot) or experimental (protein lysate) replicates? Plotting individual datapoints here would also improve transparency.

Authors: Thank you for this suggestion. We have made this change as suggested to clarify the nature of replicates. In figure 4C, we presented 4 lanes showing biological replicates (each lane represents an individual cell line). As those results were similar in terms of yield of protein across samples and treatments, we didn't perform additional technical replicates. We also modified figure 4D to better show the distribution of the data.

* In the new Fig 4E, the legend asks readers to note the shift in 3/4 cell lines. Are these the only technical replicates for each sample? Can any statistics be performed to give confidence that there is reproducibly a shift?

Authors: Thank you for bringing this point. We performed the same analysis independently twice using all four cell lines in high and low oxygen and observed the same trend, i.e., low oxygen samples with increased fluorescent intensity. As the goal of the analysis is to show a change in fluorescent intensity, meaning change in mitochondrial activity, we thought that the histogram option was the clearest way to show the difference caused by the treatment (low oxygen environment vs regular oxygen). This analysis does not generate numbers that could be used for statistical analysis, and to do that we would be creating an artificial measurement that in our opinion is not biologically justified.

* The MEF-contamination correction approach is unusual but pragmatic; it would benefit from a brief explicit statement of what exactly is subtracted or filtered rather than directing readers to Gultinan et al. (2025).

Authors: We have added this to the methodology as suggested by the reviewer.

* Trimmed Mean of M-values is a library size correction, rather than batch correction. The authors should clarify whether a separate batch correction was performed?

Authors: Thank you for the comment. We have added this paper to the methodology as suggested by the reviewer. To clarify the reviewer question, we did not perform any additional batch correction after library size correction by TMM. As shown in the PCA plot below, because samples separated primarily by the environment of culture (regular oxygen versus low oxygen) rather than batch-of-origin, no additional batch correction was applied in the dataset.

* The GEO accession number was provided to reviewers but has not been updated in the manuscript itself.

Authors: We have added this to the methodology as suggested by the reviewer.

* Some references seem to be incomplete (e.g. Yang et al. (2024) and Zhi et al. (2024) lack journal information).

Authors: We have added this to the methodology as suggested by the reviewer.

Third decision letter

MS ID#: bio.062352R2

MS Title: Low oxygen environment alters transcripts related to energy metabolism without altering the pluripotency core of bovine embryonic stem cells

Authors: Ramon C. Botigelli, Rachel Braz Arcanjo, Carly Gultinan, Justin M. Smith, Amanda J. Morton and Anna C. Denicol

I am happy to tell you that your manuscript has been accepted for publication in Biology Open, pending our standard publication integrity checks. It was accepted on 15th April 2026.